# Genetic diversity of 1,845 rhesus macaques improves genetic variation interpretation and identifies disease models

Jun Wang[1,2], Meng Wang [1,2], Ala Moshiri[3], R. Alan Harris [1,2], Muthuswamy Raveendran [1,2], Tracy Nguyen[4], Soohyun Kim[4], Laura Young [4], Keqing Wang[1,2], Roger Wiseman[5], David H. O'Connor [5], Zach Johnson[6], Melween Martinez[7], Michael J. Montague [8], Ken Sayers[9], Martha Lyke[9], Eric Vallender [10], Tim Stout[11], Yumei Li [1,2], Sara M. Thomasy [3,4,12], Jeffrey Rogers [1,2] & Rui Chen [1,2] ✉

Understanding and treating human diseases require valid animal models. Leveraging the genetic diversity in rhesus macaque populations across eight primate centers in the United States, we conduct targeted-sequencing on 1845 individuals for 374 genes linked to inherited human retinal and neurodevelopmental diseases. We identify over 47,000 single nucleotide variants, a substantial proportion of which are shared with human populations. By combining rhesus and human allele frequencies with established variant prediction methods, we develop a machine learning-based score that outperforms established methods in predicting missense variant pathogenicity. Remarkably, we find a marked number of loss-of-function variants and putative deleterious variants, which may lead to the development of rhesus disease models. Through phenotyping of macaques carrying a pathogenic OPA1:p.A8S variant, we identify a genetic model of autosomal dominant optic atrophy. Finally, we present a public website housing variant and genotype data from over two thousand rhesus macaques.

Animal models that accurately recapitulate human physiological conditions are essential for understanding diseases and developing effective treatments. However, valid animal models are still lacking for many human diseases, primarily due to the differences between humans and rodents—the most commonly used model organisms. This gap could potentially be addressed by utilizing nonhuman primates (NHPs), which are much more closely related to humans[1]. For instance, NHPs are more suitable for modeling human retinal diseases, especially macular disorders, as their retinal architecture closely resembles that of humans, including a cone-dominated macula and fovea[2], while rodents lack maculae. Similarly, NHPs provide the most translational models for studying human neurodevelopmental disorders, such as

[1]Human Genome Sequencing Center, Baylor College of Medicine, Houston, Texas, USA. [2]Department of Molecular and Human Genetics, Baylor College of Medicine, Houston, Texas, USA. [3]Department of Ophthalmology & Vision Science, School of Medicine, UC Davis, Sacramento, California, USA. [4]Department of Surgical and Radiological Sciences, School of Veterinary Medicine, University of California-Davis, Davis, California, USA. [5]Wisconsin National Primate Research Center, University of Wisconsin-Madison, Madison, Wisconsin, USA. [6]Emory National Primate Research Center, Emory University, Atlanta, Georgia, USA. [7]Caribbean Primate Research Center, University of Puerto Rico, Punta Santiago, Humacao, Puerto Rico. [8]Department of Neuroscience, Perelman School of Medicine, University of Pennsylvania, Philadelphia, Pennsylvania, USA. [9]Southwest National Primate Research Center, Texas Biomedical Research Institute, San Antonio, Texas, USA. [10]Tulane National Primate Research Center, Tulane university, Covington, Louisiana, USA. [11]Department of Ophthalmology, Cullen Eye Institute, Baylor College of Medicine, Houston, Texas, USA. [12]California National Primate Research Center, University of California-Davis, Davis, California, USA. ✉e-mail: ruichen@bcm.edu

autism spectrum disorder, due to their close similarity to human central nervous system (CNS) anatomy, physiology, and complex social behavior[3,4].

Rhesus macaques (*Macaca mulatta*), among NHPs, show a high-level genomic similarity to humans, with approximately 93% overall genome sequence similarity and greater than 97% similarity in gene coding and amino acid sequences[5]. Furthermore, disease susceptibility genes and disease risk alleles are similar between rhesus macaques and humans, making them highly valuable in studying human genetic diseases[6–10]. Extensive effort has been made to develop macaque models of human disorders, such as achromatopsia, age-related macular degeneration, Krabbe disease, autism, Batten disease, Bardet-Biedl syndrome, and Coats-like retinopathy[3,6–15]. Moreover, the mGAP [https://mgap.ohsu.edu/] online database contains searchable genotype data from genome sequencing of rhesus macaques, facilitating the potential development of macaque models[16]. However, for genetic disorders with low disease prevalence, a substantial cohort of individuals must be screened to identify individuals carrying spontaneous pathogenic variants in a given gene of interest. Additionally, although it is possible to generate genetically modified macaque models through genetic engineering, this approach is still technically demanding, labor-intensive, and costly[17,18]. To overcome these challenges, we employed a reverse genetics approach that leveraged genetic diversity in rhesus macaque populations to identify and establish NHP models for inherited human retinal diseases (IRDs) and neurodevelopmental disorders (NDs), while also improving our ability to predict the pathogenicity of genetic variants. We profiled the genetic diversity of rhesus macaque populations across eight primate research centers in the United States and uncovered rhesus macaques carrying naturally occurring pathogenic mutations (Fig. 1, Supplementary Data 1).

## Results

### Identification of single nucleotide variation in rhesus macaque populations

We conducted targeted sequencing on 286 IRD genes and 88 ND genes in 1845 individuals collected from eight primate research centers across the US representing the population diversity (Fig. 2a, Supplementary Data 1 and 2). The sequencing achieved high read coverage with an average depth of 70.2 at single-nucleotide variant (SNV) sites (Supplementary Fig. 1). After quality control and variant filtering (Supplementary Fig. 1 and 2), 47,743 SNVs were identified, averaging 2329 SNVs per individual, with 67.4% heterozygous variants and 32.6% homozygous non-reference variants. The majority of the SNVs are rare in the rhesus macaque population, with 26.0% of autosomal SNVs being singletons and 78.9% having allele frequency $\leq$ 0.5% (Fig. 2b). The transition-to-transversion ratio is approximately 3.0, aligned with a higher rate in the coding region (https://genome.sph.umich.edu/wiki/SNP_Call_Set_Properties). Notably, SNVs (33.1%) are located disproportionately in CpG dinucleotide sites, showing a 3.7-fold enrichment compared to CpG dinucleotide bases (8.9%) in the targeted regions (two-sided binomial test $p = 4.9 \times 10^{-324}$). This result suggests, as expected, that the deamination of methylated cytosines contributes to the higher mutation rate at CpG sites[17,18]. Interestingly, we observed significantly stronger purifying selection in ND genes compared to IRD genes, as indicated by a significantly lower dN/dS ratio and higher Neutrality Index (NI) for ND genes than for IRD genes (one-sided Wilcox rank sum test, $p = 1.83 \times 10^{-7}$ for dN/dS ratio and $p = 9.18 \times 10^{-5}$ for NI, Fig. 2c,d Methods)[19]. These observations are consistent with previous findings that indicate genes associated with the "Neuronal System" or autism spectrum disorder are under unusually strong negative selection[20,21].

Most (99.5%) of the polymorphic sites in rhesus macaques have orthologous positions in the human genome (hg19)[22]. Additionally, a substantial proportion of genetic variation is shared between rhesus

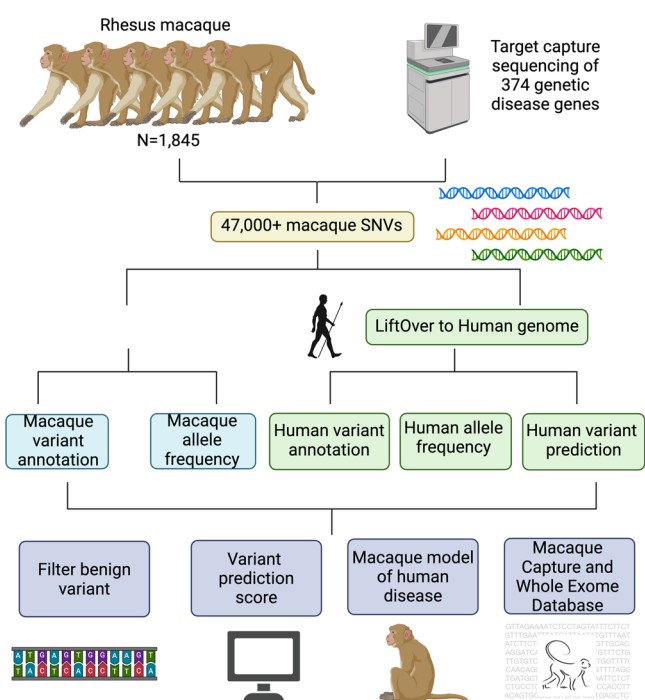

**Fig. 1 | The study design.** The schematic plot of the study design. Created with BioRender.com.

macaques and humans, including 14,933 macaque SNVs present in human populations (recorded in the Genome Aggregation Database, gnomAD [https://gnomad.broadinstitute.org/])[23] and 14,779 macaque non-reference SNV alleles corresponding to the reference alleles in the human genome (hg19). However, allele frequencies (AFs) of the shared SNVs significantly differ between macaque and human populations (Fig. 2e), with 43.9% of these sites located in CpG sites.

### A marked number of putative deleterious variants were identified

Importantly, 7131 of the macaque SNVs are predicted to affect protein-coding in both the rhesus macaque and human genomes, including 29 stop-gain and 26 splicing variants (Fig. 2f). Among the 55 loss-of-function variants, 19 stop-gain, and 24 splicing variants may cause nonsense mediated mRNA decay, likely to be pathogenic (Supplementary Data 3). Moreover, a substantial number (4622) of the SNVs were found in human disease variant databases, the Human Gene Mutation Database (HGMD) and the ClinVar [https://www.ncbi.nlm.nih.gov/clinvar/] database, with 31 SNVs classified as putative pathogenic and 2747 as likely benign (Supplementary Data 4). In total, 66 loss-of-function and reported putative pathogenic variants were identified in 45 IRD genes and 8 ND genes.

About half (30 of 66) of these putative deleterious variants were detected in multiple rhesus macaques. For example, a stop-gain variant (NM_020366:exon13:c.T1736A:p.L579X) in *RPGRIP1*, an autosomal recessive gene accounting for about 5% of Leber congenital amaurosis and also associated with cone-rod dystrophy and retinitis pigmentosa, was found in six heterozygous macaques (Supplementary Fig. 3a). RPGRIP1 exhibit species-specific colocalization with RPGR, in rod and cone outer segments in humans but in connecting cilia in mice[24]. Therefore, the macaque models could provide additional insights into the disease mechanisms. Similarly, a splice acceptor variant (NM_003560:c.1078-2 A > G) in *PLA2G6*, an autosomal recessive gene linked to Parkinson diseases and neurodegeneration, was detected in five heterozygous macaques (Supplementary Fig. 3b).

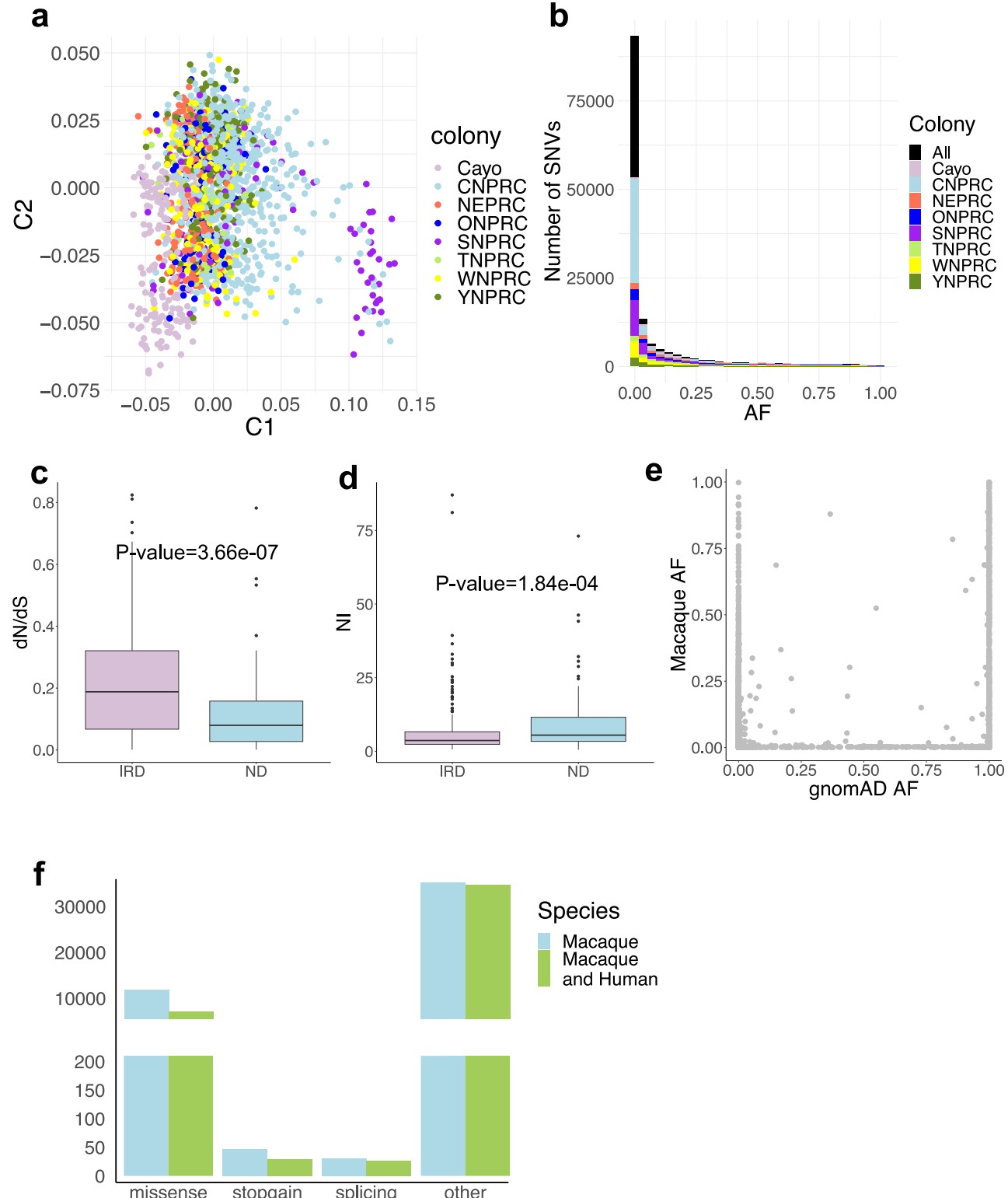

While all the putative deleterious variants were found in heterozygotes, three variants were also observed in homozygotes. For example, a stop-gain variant (NM_001292009:c.C8608T:p.R2870X) in *EYS*, a common gene for retinitis pigmentosa in autosomal recessive mode, was detected in one homozygous macaque and in 20 heterozygous macaques (Supplementary Fig. 3c).

Notably, the majority of these variants (83.3%) were exclusively found in a single colony, with a smaller proportion (16.7%) observed in multiple colonies. For instance, a known pathogenic missense variant

(NM_000350:c.G6416A:p.R2139Q) in *ABCA4*, a major gene associated with Stargardt and other macular-related diseases in autosomal recessive mode, was found in 21 heterozygous macaques across two colonies (Supplementary Fig. 3d, e). Given the relatively mild phenotype in *ABCA4* mutant mice, developing *ABCA4* rhesus macaque models could significantly advance research and development of therapeutics for Stargardt disease and related ABCA4 conditions. Overall, these variants and macaque carriers provide promising opportunities for developing rhesus models of IRDs and NDs and

**Fig. 2 | Identification of SNVs in rhesus macaque populations. a** Scatter plot showing the Multidimensional Scaling (MDS) analysis of rhesus macaque genotype data for 268 SNVs, with the 1st component represented on the $X$ axis and the 2nd component represented on the $Y$ axis. Each dot corresponds to an individual, colored by the primate center of the individual ($n = 1845$). The 268 SNVs were the autosomal variants that were sequenced in all the individuals and had allele frequency $\geq 0.01$. **b** Histogram showing the allele frequency distribution of the autosomal SNVs in each primate center and in all the centers (number of the SNVs=46,444). **c** Box plot showing the distribution of dN/dS ratio for inherited retinal disease (IRD) genes (purple, $n = 259$) and neurodevelopmental disorder (ND) genes (blue, $n = 86$). Two-sided Wilcoxon rank sum test, $P$-value = 3.66e-07. d, Box plot showing the distribution of Neutrality Index (NI) values for IRD genes (purple, $n = 259$) and ND genes (blue, $n = 86$). Two-sided Wilcoxon rank sum test, $P$-value = 1.84e-04. For box plot (**c** and **d**), the center represents the median, while the lower and upper bounds correspond to the first and third quartiles, respectively. The whiskers extend up to 1.5 times the interquartile range, and the minima

and maxima represent the observed minimum and maximum values. **e** Scatter plot showing allele frequencies of shared autosomal genetic polymorphisms ($n = 14,412$) between human ($X$ axis) and rhesus macaque ($Y$ axis) populations. The human allele frequency was obtained from The Genome Aggregation Database (gnomAD AF). The rhesus macaque allele frequency was obtained from this study (Macaque AF). Only variants with an allele number $\geq 200$ in the rhesus macaque population were considered to ensure the accuracy of allele frequency estimation. **f** Bar plot showing the numbers of SNVs categorized based on their variant consequence according to gene annotation in the rhesus macaque genome (blue) or in both the rhesus macaque and human genomes (green). The "other" category of rhesus macaque variants includes intron_variant ($n = 15,354$), synonymous_SNV ($n = 17,276$), 3_prime_UTR_variant ($n = 509$), downstream_gene_variant ($n = 1125$), upstream_gene_variant ($n = 649$), 5_prime_UTR_variant ($n = 353$), coding_sequence_variant ($n = 3$), start_lost ($n = 6$), stop_retained_variant ($n = 9$) and stop_lost ($n = 2$). Source data are provided in the Source Data file.

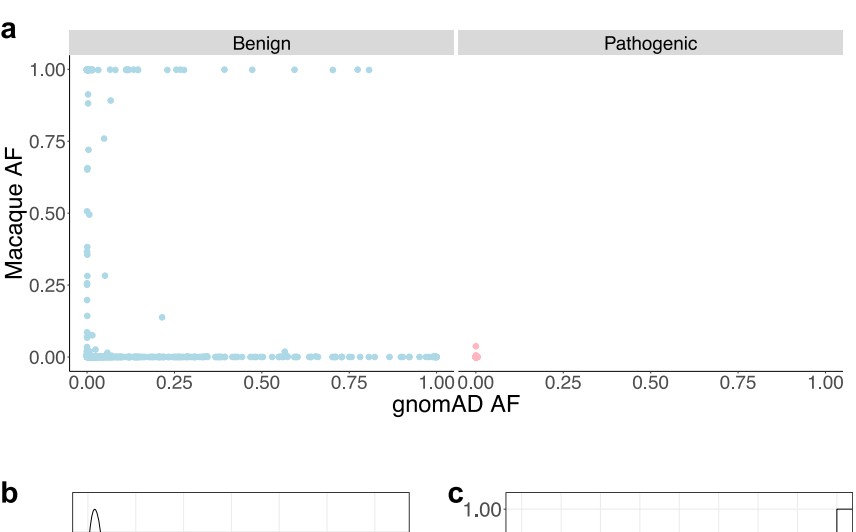

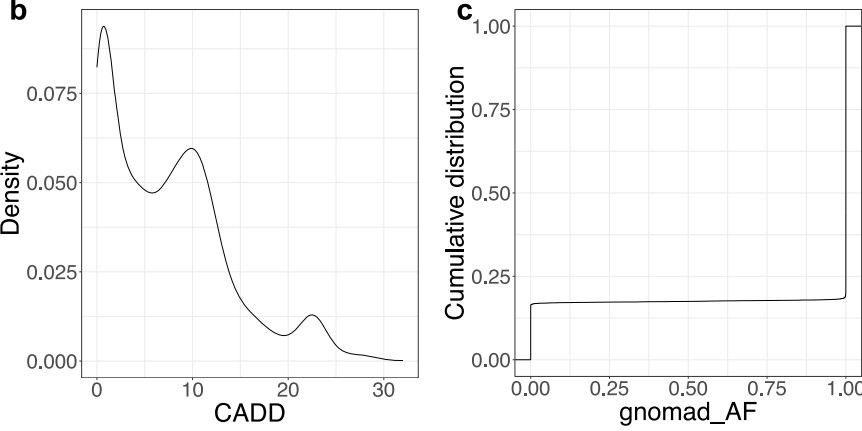

**Fig. 3 | Application of macaque allele frequency to filter putative benign variants. a** Scatter plot showing allele frequency of the reported benign variants ($n = 2919$) (blue) are significantly discordant in macaque and human populations, while allele frequency of the reported putative deleterious variants ($n = 11,208$) (pink) are consistently low in both the populations. $X$ axis indicates human allele frequency obtained from gnomAD (gnomAD AF). $Y$ axis indicates rhesus macaque allele frequency from this study (Macaque AF). **b** Density plot showing the distribution of CADD_phred score of the filtered putative benign variants ($n = 3258$). **c** The cumulative distribution of human allele frequency (obtained from gnomAD as "gnomad_AF") of the filtered putative benign variants ($n = 19,417$). Source data are provided in the Source Data file.

investigating the association between phenotype and genotype for human genetic variation.

## Filtering human benign variants with allele frequencies in macaque and human populations

Intriguingly, we observed discordant AFs for reported benign variants between rhesus macaque and human populations, in contrast to consistently low AFs for reported deleterious variants in both species (Fig. 3a, the "Filtering of benign variants" section in Methods). This led

us to identify putative benign variants using AFs in the two populations. We filtered variants with AFs higher than the maximum AF of reported deleterious variants, resulting in 19,417 macaque SNVs assigned as putative benign variants (Methods, Supplementary Data 5). These filtered variants are further supported by a low mean CADD score (7.28, $n = 3258$, Fig. 3b), ClinVar annotations (536 reported as "Benign/Likely_benign"), and absence of polymorphism in human populations (gnomAD, $n = 10,835$). Moreover, 70 filtered variants were reported as "Conflicting_interpretations_of_pathogenicity," and

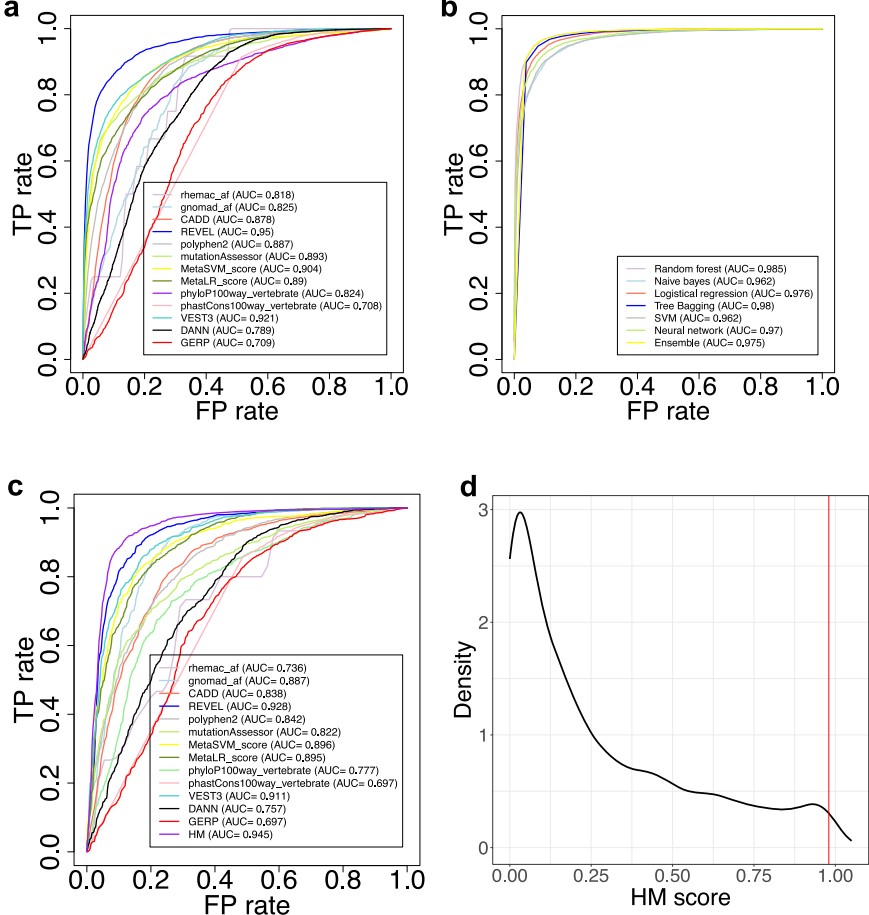

**Fig. 4 | Development of an integrative score to predict the pathogenicity of missense variants. a** The AUC distributions of 13 established in-silico scores for predicting the deleteriousness of 10,836 missense variants from ClinVar, including 5079 pathogenic variants and 5757 benign variants. TP: true positive. FP: false positive. **b** The AUC distributions of the integrative scores for assessing the 10,836 missense variants derived from seven machine learning models. **c** The AUC distributions of the integrative score based on the random forest model (the HM score) and 13 incorporated scores for assessing an independent test set from SwissVar. This test set comprised 889 benign variants and 1930 deleterious variants. **d** The distribution of the integrative HM score for 6195 missense variants in the rhesus macaque population. The red line indicates the HM score corresponding to the top 1% threshold (HM Score=0.98). Source data are provided in the Source Data file.

107 were classified as "Uncertain_significance" in ClinVar, suggesting the potential utility of this approach for annotating uncertain variants in human patients. For example, an intronic variant in *ALMS1* (NM_015120.4:c.450+11 T > C) with conflicting ClinVar interpretations, has a high rhesus macaque AF of 0.23 and homozygote frequency (HF) of 0.06 (66 of 1179), suggesting that it is likely benign. This conclusion is consistent with the spliceAI prediction (DS_AG = 0.02, DS_DG = 0.01). Similarly, a missense variant in *ABCC6* (NM_001171:c.T2237C:p.I746T), predicted deleterious by Polyphen and SIFT and classified as "Uncertain_significance" by Clinvar, has a high macaque AF of 0.38 and HF of 0.15 (280 of 1843), indicating it is likely benign. This conclusion aligns with the presence of this variant in multiple mammalian species.

Furthermore, many (*n* = 3201) filtered benign SNVs are rare in humans (human AF < 0.1% in gnomAD), suggesting this approach may be effective in identifying rare benign variants in human populations (Fig. 3c). For example, a missense variant in *FOXP1* (NM_032682:c.G64A:p.G22S), with a human AF of $4.0 \times 10^{-6}$ and ClinVar "Uncertain_significance", has a macaque AF of 0.37 and HF of 0.14 (256 of 1837), suggesting it is likely benign. Collectively, 107 human variants labeled as "Uncertain_significance" and 52 human variants labeled as "Conflicting_interpretations_of_pathogenicity" in ClinVar are rare in humans (human AF < 0.1% in gnomAD) but show a higher frequency in macaques (macaque AF > 3.8%), supporting a benign classification. These findings highlight the rhesus macaque population AF as an additional source of valuable information for annotating human variants.

## Development of an integrative score to predict pathogenicity of missense variants

Given the power of macaque and human AF data for identifying putative benign variants, we integrated them with established in-silico prediction scores to predict the pathogenicity of missense variants. We collected the training data from ClinVar (including 5079 pathogenic variants and 5757 benign variants) and aggregated 13 features (including human and macaque AFs, Polymorphism Phenotyping v2 (polyphen2) score, rare exome variant ensemble learner (REVEL) score, Combined Annotation Dependent Depletion (CADD) score, phastCons and phyloP, etc, Methods). The Area Under the Curve (AUC) value for each of the 13 features ranged from 0.708–0.95 (Fig. 4a, Supplementary Fig. 4). By combining the 13 features, we trained an integrative score using seven different machine learning models respectively (Methods). The integrative score consistently showed high AUC values (ranging from 0.962 to 0.985) across the seven models, with the random forest model achieving the best performance (AUC = 0.985) (Fig. 4b). We referred to the integrative score derived from the random forest model as "Human Macaque Score" (HM score). The HM score, when applied to an independent test set from SwissVar (889 benign variants and 1930 deleterious variants, Methods),

achieved an AUC of 0.945, outperforming each individual score incorporated into the model (Fig. 4c).

We then calculated the HM score for 6195 rhesus macaque missense SNVs (Supplementary Data 6), and assigned variants ranking in the top 1% (HM score ≥ 0.98) as potentially deleterious (Fig. 4d, Supplementary Fig. 5). Consequently, we identified 68 putative deleterious missense variants in 49 IRD genes and 8 ND genes from 274 macaques. Notably, some of the identified putative deleterious variants were found in haploinsufficiency disease genes in autosomal dominant inheritance mode, such as *ITM2B*, *JAG1*, *KIF11*, *LRP5*, *NR2F1*, *PRPF31*, *SNRNP200*, *VCAN*. This suggests that the rhesus macaque carriers of the identified putative deleterious variants can be directly examined to confirm the phenotype. Additionally, some genes in autosomal recessive inheritance have multiple putative deleterious variants identified, for example, *CLN3*, *CNNM4*, *ABCA4*, *CC2D2A*, *EYS*. This provides potential breeding candidates to develop rhesus macaque disease models. Overall, these results offer priority for future analyses and the development of rhesus macaque models of diseases.

### Identification of a rhesus macaque model of human optic atrophy

Remarkably, among the shared SNVs between macaques and humans, a missense variant in *OPA1* (NM_015560.2:c.22 G > T:p.ala8ser) was previously reported in patients with autosomal-dominant optic atrophy (ADOA)[25] (https://www.ncbi.nlm.nih.gov/ClinVar/variation/193386/?new_evidence=false). This variant is in the first exon and leads to the substitution of alanine with serine within the mitochondrial targeting signal region of the OPA1 protein. It is predicted to be moderately deleterious (a REVEL score of 0.622, a CADD PHRED score (v1.6) of 23, a HM score of 0.83, and predicted as "Damaging" by Sorting Intolerant From Tolerant (SIFT), etc). This change leads to the formation of a new hydrogen bond between OPA1.p8S and OPA1.p7A (Supplementary Fig. 6a), and affects a nucleotide that is conserved across vertebrates (a phyloP100way_vertebrate score of 0.663 and a phastCons100way_vertebrate score of 0.932, Supplementary Fig. 6b). This variant is absent in the general human populations (gnomAD) and has a low macaque AF of 0.019, suggesting it is rare in both species. Taken together, these findings suggest this variant is a putative pathogenic variant associated with ADOA.

To validate the above hypothesis, we conducted phenotyping on heterozygous rhesus macaques carrying this mutation and compared them to age-, sex-matched rhesus macaque controls. The most severely affected macaque was a 28-year-old female who displayed marked optic nerve head atrophy and significantly reduced peripapillary retinal nerve fiber layer (RNFL) thickness across nearly all regions in comparison to age-, sex-matched control macaque (Fig. 5a–g, and Supplementary Fig. 6c). Furthermore, a significant decrease was observed in peripapillary RNFL thickness in the superotemporal region of eight heterozygous macaques carrying the mutation compared to eight age-, sex-matched control macaques (Fig. 5h, and Supplementary Data 7), similar to what is seen in humans with OPA1 mutations. These findings indicate that we have identified a rhesus macaque genetic model of ADOA.

### An online database with searchable variants and genotype data of over two thousand rhesus macaques

To make all the data from this study publicly accessible, we developed the Macaque Capture and Whole Exome Database (mCED) [https://ird.research.bcm.edu/macaque/] browser (the web link can also be found at https://rchenlab.github.io/resources/). This web-based interface allows users to interactively explore and download the dataset. mCED contains variant annotation and genotype information of 1408 rhesus macaques through targeted sequencing, 615 rhesus macaques via whole-exome sequencing, and 446 rhesus macaques through both targeted and whole-exome sequencing (The data of whole-exome sequencing will be addressed in another paper).

Users can perform queries through various search criteria, including gene name, variant position, and genomic regions. Each gene has a summary page that provides detailed information and resources, such as links to other browsers (e.g., UCSC Browser, Ensembl, GeneCards, OMIM, PubMed Search, Wikigenes, GTEx, etc), and includes a list of variants identified within the gene locus (Fig. 6a). Variants can be filtered based on variant types (e.g., missense, synonymous) according to macaque or human gene annotation. Furthermore, each variant is linked to a summary page containing comprehensive information, including allele frequency, variant effect annotation, links to external resources (e.g., dbSNV, UCSC, ClinVar, NHLBI ESP, and gnomAD), and in silico prediction scores of the corresponding human variant (e.g., SIFT, Polyphen2, REVEL, CADD and etc.) (Fig. 6b). Additionally, for each variant, information about the individuals carrying that variant is provided, including genotype and colony details. This information enables users to contact the corresponding primate center for further inquiries and follow-up studies (Fig. 6b).

## Discussion

The lack of valid animal models for human diseases poses a significant bottleneck in biomedical research. This study addresses this challenge by employing a reverse genetic approach, sequencing spontaneous genetic variations in 1845 rhesus macaques across eight primate centers in the US. Over 47,000 high-quality rhesus macaque SNVs were identified, characterized by high sequencing read coverage and stringent QC filtering. A substantial proportion of the genetic variations are shared between macaques and humans, enabling translational studies of these genetic variants and the macaque carriers. By analyzing AFs in both macaques and humans, we identified putative benign variants, addressing limitations in annotating rare benign and uncertain variants in humans solely based on human data. Through integrating macaque and human data with in-silico scores, we developed a machine learning-based integrative score to predict the pathogenicity of missense variants, outperforming established prediction methods. Moreover, we identified 66 loss-of-function and reported putative pathogenic variants and 61 putative deleterious missense variants in the macaques, generating promising opportunities to develop models for human IRDs and NDs. To establish macaque models, macaque carriers of deleterious variants should be validated using Sanger sequencing, and additional carriers can be identified through pedigree tracing. Further phenotyping of individuals carrying homozygous recessive variants or heterozygous dominant variants, as well as positive outcomes from breeding carriers of recessive variants, can be utilized to develop rhesus macaque models of IRDs and NDs[6].

With this reverse genetics approach, we identified a genetic rhesus macaque model of human ADOA, carrying a naturally-occurring putative pathogenic mutation in *OPA1* previously reported in human patients[25]. Rhesus macaques exhibit a spectrum of phenotypic severity similar to human ADOA patients[26]. Phenotypic severity appeared to be age-dependent with the most severe phenotype observed in the oldest animal examined, a 28 year-old female. Studies are ongoing to define the disease onset and progression in this NHP model of ADOA.

To make this resource publicly accessible, we created the mCED browser, consolidating variant and genotype data from over two thousand rhesus macaques through targeted and whole exome sequencing. This interactive interface enables comprehensive data exploration and download, offering information for inquiries, subsequent studies on potential rhesus macaque models, and aiding in assessing variant pathogenicity in humans or other model organisms. The variant information can also guide the design of PCR primers and SNV arrays for studies on the rhesus macaque genome. One potential enhancement is to integrate the phenotype data of the sequenced

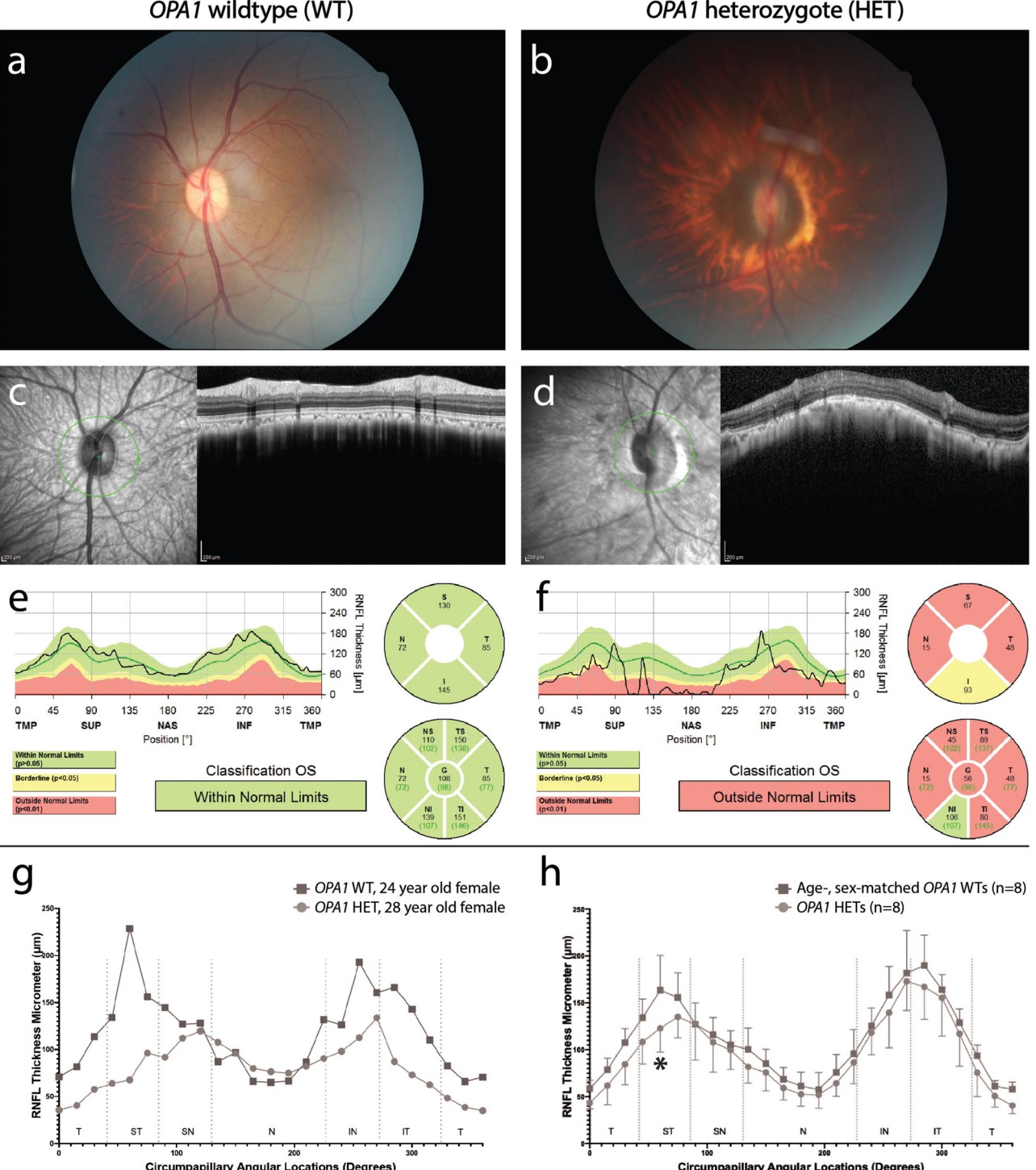

**Fig. 5 | A genetic rhesus macaque model of human optic atrophy.** Rhesus macaques heterozygous for an optic atrophy 1 (OPA1) mutation exhibit structural and functional changes consistent with retinal ganglion cell (RGC) loss. The horizontal ovoid, light pink optic nerve head of a normal 24-year-old female rhesus macaque wildtype for the OPA1 mutation (**a**) differs from the markedly smaller, atrophic optic nerve head of a 28-year-old female heterozygous for an OPA1 mutation (**b**); the OPA1 heterozygote did have moderate cataract which limited the quality of the fundus photography. Automated peripapillary optic nerve head scans of the normal control (wildtype: WT) (**c** and **e**) demonstrate normal retinal nerve fiber layer (RNFL) thickness in all regions while the OPA1 heterozygote (HET) (**d** and **f**) demonstrates a markedly reduced RNFL thickness in essentially all regions.

TMP (T): Temporal, SUP (S): Superior, NAS (N): Nasal, INF (I): Inferior, ST: Superotemporal, IT: Inferotemporal, SN: Superonasal, IN: Inferonasal, and OS: left eye. Manual RNFL measurements of the two rhesus macaques (**g**) confirmed the findings in the automated scans. Manual RNFL measurements were compared between 8 OPA1 heterozygotes and 8 age-, sex-matched wild type controls (Supplementary Data 7) using a two-way analysis of variance (ANOVA) and Sidak's multiple comparison test. Peripapillary RNFL was significantly decreased (*$P = 0.03$) in the superotemporal region in 8 OPA1 heterozygotes versus 8 age-, sex-matched controls (**h**). Data represent mean values ± standard deviation. Source data are provided in the Source Data file.

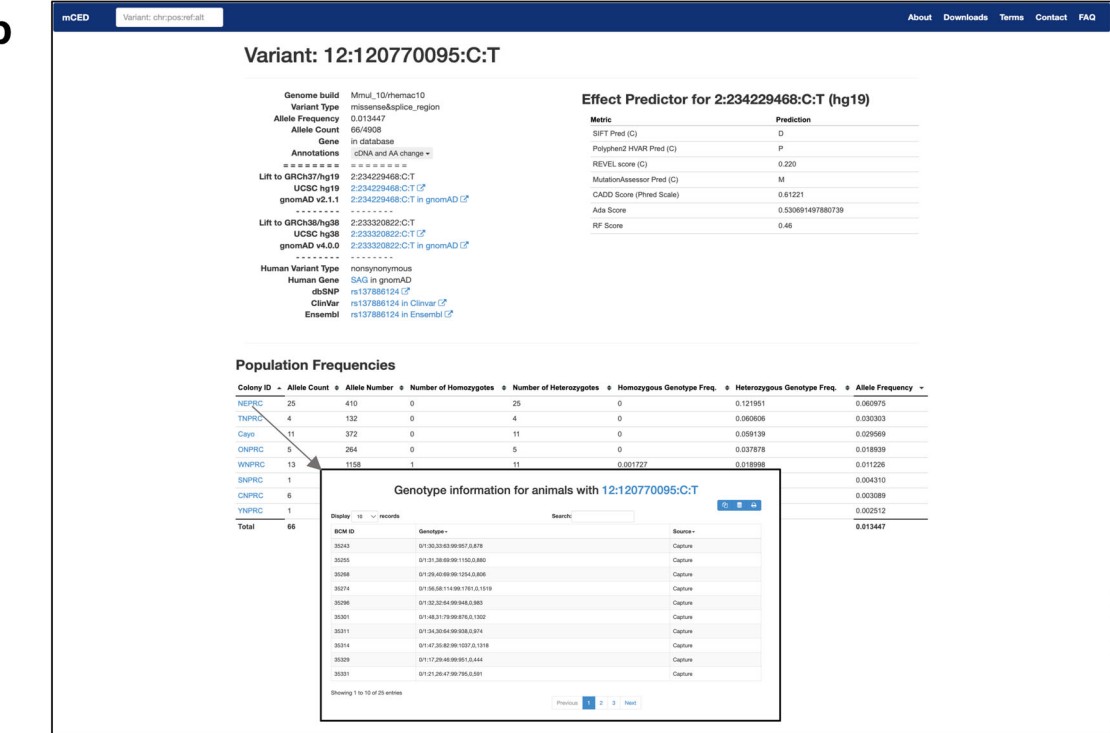

**Fig. 6 | The Macaque Capture and Whole Exome Database (mCED) browser.** **a** The gene page lists detailed information about a gene and the rhesus macaque genetic variants in that gene locus. **b** The variant page lists comprehensive information regarding a variant, including allele frequency, variant annotation, links to popular genome browsers, and established in silico prediction scores of the corresponding human variant.

individuals into the mCED browser. This addition will facilitate the study of genotype-phenotype relationships and the development of animal models.

In conclusion, the animal model, data, tools, and methodology developed in this study enhance our understanding of genetic variation in human diseases and facilitate translational studies with NHP

models, paving the way for developing treatments. Our work provides an invaluable resource for genetic, evolution, and biomedical research.

## Methods

We confirm that our research complies with all relevant ethical regulations. The rhesus macaques in this study are accredited by the Association for Assessment and Accreditation of Laboratory Animal Care (AAALAC) International. This study was conducted in accordance with the Guidelines of the Association for Research in Vision and Ophthalmology Statement for the Use of Animals in Ophthalmic and Vision Research and the National Institutes of Health (NIH) Guide for the Care and Use of Laboratory Animals. Phenotyping and ophthalmic examinations were performed according to an animal protocol approved by the UC Davis Institutional Animal Care and Use Committee.

### Animal sample collection

All animals included in this study were rhesus macaques (*Macaca mulatta*) born and maintained at the eight primate research centers in the United States (Supplementary Data 1).

### Retinal disease and neurodevelopmental disease gene panel selection

The inherited retinal disease genes in the panel were mainly collected from the RetNet database (https://web.sph.uth.edu/RetNet/) and the OMIM database (https://www.omim.org). The neurodevelopmental disease genes were primarily identified for inclusion in the panel based on the Simons Foundation Autism Research Initiative (SFARI) Gene list. SFARI Category 1 High Confidence genes and Category 2 Strong Candidate genes were included[27].

### Targeted genome sequencing

Covaris was used to shear 1 ug genomic DNA for 70 seconds and Ampure XP beads were used for the purification. Following end repair and A-tailing, the product was added with indexed adapters. Ampure XP beads were used to purify the product, and KAPA Hifi HotStart ready mix was used for amplification. 40-50 libraries were pooled for Agilent SureSelect Target enrichment system (Agilent) following the manufacturer's protocol with modification. Briefly, pooled samples were hybridized to probe pool, and captured with Dyanbeads MyOne Streptavidin T1 magnetic beads. After washing, captured DNA was amplified with Phusion High Fidelity DNA polymerase (NEB). After cleanup and quantification, the diluted library was sequenced in an Illumina Novaseq 6000 Sequencer.

### Sequencing data processing

The sequencing reads were aligned to the rhesus reference genome assembly (Mmul_8.0.1 or Mmul_10) with BWA mem (0.7.12-r1039). We followed the GATK (3.3-0-g37228af)[28] pipeline to call the single nucleotide variants (SNVs) and short insertion/deletions (indels). The variants calling based on Mmul_8.0.1 genome assembly were further mapped to the Mmul_10 assembly using UCSC genome browser program liftOver[22]. The sites with mismatched reference nucleotides were corrected with bcftools[29].

### Quality Control

QC and filtering of the sequenced variants were performed with the following criteria: 1) Indicated as "PASS" by GATK (4.0.0.0)[28] based on the filtering criteria: $QD < 5.0$, $QUAL < 30.0$, $FS > 15.0$, $MQ < 50.0$, $MQRankSum < -12.5$, and $ReadPosRankSum < -8.0$. 2) Remained dimorphic after assigning the variants as missing that matched any of the following criteria: a) the variants not called; b) heterozygous variant calls with allelic imbalance ($AB > 0.8$ or $AB < 0.2$); c) variant GQ per sample (from GATK) < 20; d) genotype was supported by less than 10 reads ($DP < 10$). 3) the sites where more than two distinct alleles are

detected in a single subject were filtered. The samples with sequencing coverage less than 10x were filtered out. The QC and filtering were performed using GATK and custom scripts in Perl[16,28] (Supplementary Fig. 1, 2).

### The origin of the rhesus macaques included in this study

All rhesus macaques in this study are housed in the primate research centers. The U.S. research colonies of rhesus macaques were primarily established using animals imported from India decades ago, although a much smaller number of Chinese-origin rhesus macaques have been introduced to some colonies over time. To estimate the origins of the U.S. research rhesus macaques in this study, we conducted principal component analysis (PCA) of their genotype data. We performed principal component analysis (PCA) using SNV genotypes from the 1845 capture samples together with 3 wild-caught Chinese rhesus samples (SRA Sample Accessions: SAMN03264758. SAMN03264759, SAMN03264760). Autosomal SNVs were filtered with PLINK (v1.90) for missing call rates >0.05 (--geno 0.05) and minor allele frequency <0.1 (--maf 0.1). PLINK --pca was performed on this dataset to generate eigenvectors, and principal components 1 and 2 were plotted using the R plot function.

As shown in Supplementary Fig. 7, the three wild-caught Chinese rhesus samples clustered together with a low value on PC1. Since the genotype data primarily covers the coding regions of 374 genes through targeted sequencing rather than the entire genome, the delineation of the Chinese cluster is not as clear. However, the PCA result indicates that the wild-caught Chinese samples cluster with a few samples from CNPRC and SNPRC. This result aligns with the knowledge that both CNPRC and SNPRC house a few known Chinese and Chinese/Indian hybrid animals.

Additionally, we performed Multidimensional Scaling analysis (MDS) on the genotype of the 1845 capture samples alone (Fig. 2a). The variants were filtered to keep autosomal variants that were sequenced in all the individuals and have allele frequency ≥ 0.01. The MDS was implemented with PLINK v1.90b5.2.

### Molecular evolution analysis

To compute Ka/Ks ratio, the primary transcript ID (the longest coding transcript) per gene was obtained for the Macaca mulatta and Homo sapiens genomes respectively, downloaded from Ensembl: https://ftp.ensembl.org/pub/release-101/gff3/macaca_mulatta/Macaca_mulatta.Mmul_10.101.gff3.gz and https://ftp.ensembl.org/pub/grch37/release-87/gff3/homo_sapiens/Homo_sapiens.GRCh37.87.gff3.gz. The coding sequences of the primary transcripts were obtained from https://ftp.ensembl.org/pub/release-101/fasta/macaca_mulatta/cds/Macaca_mulatta.Mmul_10.cds.all.fa.gz and https://ftp.ensembl.org/pub/grch37/release-87/fasta/homo_sapiens/cds/Homo_sapiens.GRCh37.cds.all.fa.gz, and were aligned using macse_v0.9b1.jar in codon-aware manner[30]. Subsequently, the pair-wise aligned coding sequences underwent Ka/Ks calculation using the "codeml" program from the "paml4.9j" package[31]. Additionally, based on the pair-wise aligned coding sequences and the aligned amino acid sequences output from macse_v0.9b1.jar, we calculated dN/dS. Based on the missense/stop-gain variants and synonymous variants from gnomAD (gnomad.exomes.r2.1.1.sites.vcf.bgz) and this study, we calculated pN/pS. The Neutrality Index was calculated as $NI = (pN/pS)/(dN/dS)$.

### Variant annotation

The rhesus macaque variants in the rhesus macaque genome Mmul_10 assembly were annotated using the variant effect predictor[32] (VEP) in release 101, based on merged Ensembl and RefSeq gene models of Mmul_10. In addition, the rhesus macaque variants in Mmul_10 coordinates were mapped to the human genome hg19 assembly using UCSC genome browser program liftOver[22]. The sites with mismatched reference nucleotides were annotated with bcftools[29]. To predict the

protein-altering effects in the human genome, the variants in hg19 coordinates were annotated with ANNOVAR[33] (v. 07/17/2017) and dbNSFP[34,35] (v.3.5a, including SIFT, PolyPhen-2, etc.) based on the gene model of hg19. The allele frequency in the rhesus population was computed by considering the number of sequenced samples per base, while the allele frequency in the human population was annotated based on the gnomAD database (v2.1.1). The variants with the same amino acid change consequence in both macaques and humans were screened against the variants in the HGMD (v.12-20-2016) and ClinVar (v20230710) databases to identify the overlapped variants.

### Filtering of benign variants

We collected the variants in the HGMD (v.12-20-2016), ClinVar (v20230710) and/or SwissVar [https://ftp.uniprot.org/pub/databases/uniprot/current_release/knowledgebase/complete/docs/humsavar.txt] databases, and selected those with orthologous variants in the rhesus macaque genome and targeted in our sequencing. Among them, we assigned putative deleterious variants as follows: variants labeled as "Disease_causing_mutation" in HGMD that are not annotated as benign in ClinVar and SwissVar, variants labeled as "Pathogenic", "Likely_pathogenic", or "Pathogenic/Likely_pathogenic" in ClinVar that are not annotated as benign in SwissVar, and variants labeled as "LP/P" in SwissVar that are not annotated as benign in ClinVar. In parallel, we defined likely benign alleles as those labeled as "Benign", "Likely_benign" or "Benign/Likely_benign" in ClinVar and "LB/B" in SwissVar. As a result, we identified a total of 11,208 putative deleterious variants and 2919 putative benign variants. To filter out benign variants, we adopted a conservative approach by selecting the allele frequency filtering threshold based on the highest allele frequency of these 11,208 putative deleterious variants. This method might include some false positives of deleterious variants, but it minimizes the risk of filtering out true pathogenic variants. Specifically, we took the maximum allele frequency of the 11,208 putative deleterious variants in rhesus macaque (AF = 0.038) and human (AF = $5.1 \times 10^{-3}$, based on gnomAD) populations as the cutoff to filter the likely benign variants sequenced. If a variant had AF higher than either of the two maximum AFs in the corresponding species, it was classified as a putative benign variant.

### Identification of putative pathogenic variants

To identify the reported putative pathogenic variants, we collected the "Pathogenic", "Likely_pathogenic", or "Pathogenic/Likely_pathogenic" variants in ClinVar (v20230710) and the "Disease_causing_mutation" variant in HGMD (v.12-20-2016) that are not annotated as benign in ClinVar. Then we filtered out the variants that matched any of the following criteria: 1) with CADD score < 25; and 2) in the *RPGR* gene (as there might be CNV in the RPGR region). After filtering, 31 variants were left and listed in Supplementary Data 4. Sanger sequencing was conducted to confirm the selected rhesus macaques carrying putative pathogenic variants. The primers for sanger sequencing are listed in the Supplementary Data 8.

### Training machine learning models to predict the pathogenicity of missense variants

We gathered 5079 likely pathogenic missense variants and 5757 likely benign missense variants classified in ClinVar (v20230710) as the training data, which have the same amino acid change effect in both the rhesus macaque and human genomes. We collected 13 features of these variants to build a machine learning-based integrative score, including the allele frequencies in rhesus macaque and human populations, CADD phred score (v1.6)[36], revel score, polyphen2_HVAR score, mutation_Assessor score, MetaSVM score, MetaLR score, phyloP100way_vertebrate, phastCons100way_vertebrate, VEST3 score, DANN score, and GERP score from dbNSFP[34,35] (v.3.5a). We tested seven machining learning models to train the score respectively, including random forest, naïve Bayes, logistical regression, tree bagging, SVM,

neural network, and ensemble using caret_6.0-93 R package. Five-fold cross-validation was used to estimate the AUC values of the seven algorithms with cvAUC_1.1.4 R package. Among the seven models, random forest has the best performance and was used to develop the HM score. An independent test dataset was collected from SwissVar (https://ftp.uniprot.org/pub/databases/uniprot/current_release/knowledgebase/complete/docs/humsavar.txt). Only the variants that have the same amino acid change effect in both the rhesus macaque and human genomes and do not overlap with the training data were retained as the test dataset.

### Phenotyping and identification macaque model of ADOA

Sedation, ocular examinations, fundus photography, and spectral-domain optical coherence tomography (SD-OCT) with confocal scanning laser ophthalmoscopy (cSLO) were performed as previously described[37]. Briefly, sedation was induced with an intramuscular injection of ketamine hydrochloride and dexmedetomidine with or without midazolam under the monitoring of a trained technician and a veterinarian. Ophthalmic examination included external and portable slit lamp examination, as well as dilated (tropicamide 1%, phenylephrine 2.5%, and cyclopentolate 1%) indirect ophthalmoscopy[37]. Fundus photographs centered on the optic nerve were acquired with a CF-1 Retinal Camera with a 50° wide angle lens (Canon, Tokyo, Japan). The Spectralis HRA + OCT (Heidelberg, Germany) was used to perform SD-OCT with cSLO. Images and measurements of the peripapillary RNFL were obtained using a standard 12-degree diameter circular B-scan, with 1536 A-scans, centered on the ONH; three scans were acquired per eye.

### Developing the database browser

The database browser was built primarily with open-source tools and public resources, written in HTML, PHP, and SQL. The backbone was created based on the ExAC browser[38], Bootstrap version 3.1.1 (https://github.com/twbs/bootstrap) and JQuery version 2.1.4 (https://jquery.com/). All variants and metadata were loaded into MySQL Ver 8.0.20. The major components loaded include the gene data, the annotated variant data, the variants liftover to hg19, the genotype information of the animals, and pre-calculated allele frequency data. The browser contains an IGV track showing locations of variants in the Mmul_10 genome implemented by igv.js (2.15.11)[39] All released data can be easily exported and processed with existing variant-analysis tools. The entire system runs on a Redhat Linux virtual machine with 4 cores, 32 GB RAM, and 200 GB of disk space using Apache 2.4.6.

### Statistics and reproducibility

All statistical analyses performed on the data are indicated in the Methods section or figure captions. No statistical method was used to predetermine the sample size. The samples with sequencing coverage less than 10x were excluded from the analyses (see "Quality Control" section in the Methods). For phenotyping analyses of the OPA1 heterozygous rhesus macaques, we collected OPA1 heterozygous rhesus macaques and age-, sex-matched OPA1 wild-type rhesus macaques to control covariates. The Investigators were not blinded to allocation during experiments and outcome assessment. For phenotyping analyses of the OPA1 heterozygous rhesus macaques, the researchers knew the genotype when they conducted manual measurements. However, automated measurements were generated by the software.

### Reporting summary

Further information on research design is available in the Nature Portfolio Reporting Summary linked to this article.

## Data availability

The rhesus macaque genome assembly Mmul_8.0.1 [https://www.ncbi.nlm.nih.gov/datasets/genome/GCF_000772875.2/] and Mmul_10 [https://www.ncbi.nlm.nih.gov/datasets/genome/GCF_003339765.1/]

used in this study are available from the NCBI. The human genome assembly hg19 [https://hgdownload.soe.ucsc.edu/goldenPath/hg19/bigZips/hg19.fa.gz] used in this study is available in UCSC genome browser. The genome sequences of three wild caught Chinese rhesus samples used in this study are available in the Sequence Read Archive (SRA) under the sample accession numbers: SAMN03264758, SAMN03264759 and SAMN03264760[40]. The mCED browser is accessible at https://ird.research.bcm.edu/macaque/, or https://rchenlab.github.io/resources/, The sequencing data generated in this study (i.e., bam files) have been deposited in the SRA under the accession number PRJNA1107273. Source data are provided in this paper.

## Code availability

The code used for this project can be found in GitHub repository: https://github.com/fe4960/rhemac_retcap. A copy of the code version used for this publication is available from Zenodo (https://doi.org/10.5281/zenodo.11166726)[41].

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

## Acknowledgements

The work was supported by U24EY029904 to RC, JR, AM, and ST and by EY034123 to AM. Next-generation sequencing was partially supported by S10OD032189, RRF to RC, and the genomic module of P30EY002520 to SW. Phenotyping was also supported by P30EY12576 and the California National Primate Research Center Base Grant from the National Institutes of Health, Office of the Director, OD011107. This work was supported by and unrestricted grant by the Research to Prevent Blindness to the Ophthalmology Department at BCM. This research was supported in part by contract 75N93021C00006 from the National Institute of Allergy and Infectious Disease, NIH to DHO as well as the Office of Research Infrastructure Programs/OD (P51OD011106) awarded to the Wisconsin National Primate Research Center at the University of Wisconsin-Madison. The samples from the Caribbean Primate Research Center (CPRC) were supported by P40OD012217 to CPRC, R01MH096875 and R01MH089484. Additionally, we acknowledge the computing cluster server in the Department of Molecular and Human Genetics and Human Genome Sequencing Center at Baylor College of Medicine for providing the computing resource. Figure 1 was created with BioRender.com.

## Author contributions

R.C. J.R., J.W., and S.T. conceptualized and designed the study. R.C. J.R. and S.T. supervised the work. S.T., R.W., D.O., Z.J., M.M., M.J.M., K.S., M.L., and E.V. provided rhesus macaque samples. Y.L. generated NGS data. J.W., R.A.H., and M.R. designed the gene capture panel and analyzed the data. M.W. and J.W. constructed the mCED browser. S.T., A.M., T.N., S.K., and L.Y. performed the phenotyping study of rhesus macaques. J.T.S. advised the phenotyping study and disease gene collection. K.W. and J.W. performed Sanger-sequencing and analysis. J.W. wrote the first draft of the manuscript. All authors edited the manuscript and contributed to critical revisions of the manuscript.

## Competing interests

The authors declare no competing interests.
