## [Peer Review File · Nature Communications]

REVIEWER COMMENTS

Reviewer #1 (Remarks to the Author):

I thank the authors for a lovely manuscript and contribution to publicly available code and data.

The methodology and conclusions are sound and well written. The work deserves a spotlight as the rhesus macaque model is unmatched.

My only comment is that the figures could use some work for such a high impact journal.

Figure 1 is not of a quality that I expect for Nature. I would suggest it either gets deleted entirely as it isn't needed or the flow diagram be made more clear. The arrows are too large, the images too generic. I would love to see this work have a nice graphical abstract like this - but it doesn't meet the bar.

Figure 4 - the legend shouldn't be crossed by the data. Please revise.

Figure 6 is important information but the figure panels as laid out make the images far too small to be useful. This should either be split or reimagined to show the important components without being so small.

Why do all images use a different color palette? Can this be fixed to appear as a cohesive set of data?

These things do not require content change - but I would suggest that the authors would have better uptake of their manuscript with some attention to the appearance given the impact of the work.

Reviewer #1 (Remarks on code availability):

The data and database made available is excellent. All files were accessible.

Reviewer #2 (Remarks to the Author):

This is an interesting study comparing macaque sequence data from targeted sequencing of coding regions for a selection of human inherited retinal dystrophy genes (IRDs) and neurodevelopmental genes. Comparison of allele frequencies for macaque and human provides useful information for human variants, particularly those of uncertain significance. Additionally, one example of a compelling macaque model of an inherited optic nerve disease was presented. This is a well reasoned study using appropriate methods and with interesting results. The manuscript is also well written. Several comments for the authors to consider:

- 1) The primary limitation of this work is absence of phenotype for the sequenced individuals. The work would be more compelling and impactful if phenotype data was also available. Any plans to do this? Perhaps for the subcohort that underwent exome sequencing?
- 2) For the RPGRIP1 variant it would be helpful for the reader to make clear that this gene is on the X chromosome and is a cause of X-linked disease in humans. Also it would be interesting to know if the carriers of this variant are males or females. Similarly, for the PLA2G6 gene it would be helpful to note that this is a cause of autosomal recessive disease.
- 3) The data for human low frequency variants of unknown significance that exhibit higher frequency in macaques is clearly useful to support a benign classification, and several examples are provided showing this. It would be helpful to know how many (or percent) of human VUSs are in this category. This information may be included in the figures but it is a bit difficult to tease out and as this is a significant point of the manuscript it would be helpful to have in text.
- 4) In figure 2F what is included in the 'other' category? Synonymous variants? Intronic? Stop loss? UTR variants? As this appears to be the major category it would be helpful to know what is included.

Reviewer #3 (Remarks to the Author):

In their manuscript entitled "Genetic diversity of 1,845 rhesus macaques improves genetic variation interpretation and identifies macaque models for inherited human retinal and neurodevelopment diseases," Wang et al. generate a catalogue of variants using a targeted sequencing approach for 374 neurodevelopmental and retinal disease genes and present their findings. With the reverse genetic approach – screening of 1,845 of rhesus macaques from different primate research centers

for single nucleotide variants (SNVs) – the authors identify more than 47,000 SNVs. In their comparison of the rhesus macaque variants to human genetic variation databases, the authors observe that common variants in rhesus macaque that are also present in human are likely polymorphic and that variants that are rare in both species are more likely disease associated. In addition, to identify some likely pathogenic variants, the authors investigate one disease (autosomal dominant optic atrophy) in more detail and show that rhesus macaque may serve as a suitable disease model. Furthermore, Wang et al. developed a machine learning-based prediction model and a searchable database. The intersection with human variants to decipher (likely) benign from pathogenic variants is a particular strength. The prediction model and searchable database will likely be of particular interest to the human genetics community.

The combination of approaches, tools, and databases included in the manuscript is unique, especially given that the database seems to include additional genes beyond the 374 neurodevelopmental and retinal disease genes. The different directions of the manuscript may attract a broader audience. At the same time, I feel that the transitions could be reemphasized to make the manuscript more cohesive.

I had some difficulty determining the genes included in the searchable database, as some genes (e.g., genes within the MHC locus) did not provide a match in the database. Wang et al. state that there is a larger/different dataset that will be subject to another publication. However, some greater detailed information about the database and its content may be useful for readers interested in utilizing the database. For example, could a gene list (or list of excluded genes) be provided?

Related to the database, would it be possible to provide the data for different human genome assemblies (hg38 and/or the T2T assembly) to provide access to a broader community?

Wang et al. did not discuss the origin of the rhesus macaque included in this study in their manuscript. More specifically, are all rhesus macaque housed in the primate centers and included in this study of Indian origin? Or were Chinese rhesus macaques also included? If Chinese rhesus macaque were included, did the different rhesus macaque population show varying allele frequency/disease allele distribution?

The genetic diversity within rhesus macaque populations in different primate centers can vary dramatically. Did the authors observe differences between different primate centers? Did some populations show homogenization/signs for inbreeding? How did the authors account for variation between different primate centers in their model building? Are the observed allele frequencies in this study aligned with allele frequency distribution in wild rhesus macaque populations? Related to this, is the number of pathogenic variants aligned with expectations based on population genetic

analyses? Or did the authors see enrichment/depletion of pathogenic variants between primate centers compared to wild-caught rhesus macaques?

In their abstract, Wang et al. state the identification of more than 47,000 high quality SNV. The legend of Figure 2b indicates 46,444 variants overall. What accounts for the discrepancy?

The scale of the Y axis in Figure 2b appears to be off. I find it difficult to determine the actual allele frequency of the different variants on the X-axis. For example, where is .25 exactly located on the X axis? Why is there a white space within the first bar (0.00)? To what does all refer?

Is there a reason why the colors used for the different primate centers are not kept the same between Figure 2a and 2b?

The number of pathogenic variants in rhesus macaque (Figure 3) seems rather limited. Would it be possible to include a larger dataset (more genes/variants) to further support the findings?

A deeper engagement with the putative deleterious variants may be of interest to a broader research community.

Overall, I found figure 6 less informative and essential for the manuscript.

REVIEWER COMMENTS

Reviewer #1 (Remarks to the Author):

I thank the authors for a lovely manuscript and contribution to publicly available code and data.

The methodology and conclusions are sound and well written. The work deserves a spotlight as the rhesus macaque model is unmatched.

My only comment is that the figures could use some work for such a high impact journal.

Response:

We are grateful for the encouraging and positive comments by Reviewer #1.

Figure 1 is not of a quality that I expect for Nature. I would suggest it either gets deleted entirely as it isn't needed or the flow diagram be made more clear. The arrows are too large, the images too generic. I would love to see this work have a nice graphical abstract like this - but it doesn't meet the bar.

Response:

Thanks Reviewer #1 for the constructive comment. We modified Figure 1 to make the flow diagram clearer and more informative.

Figure 4 - the legend shouldn't be crossed by the data. Please revise.

Response:

Thanks Reviewer #1 for the constructive comment. We modified the legend of Figure 4 and made it not crossed by the data.

Figure 6 is important information but the figure panels as laid out make the images far too small to be useful. This should either be split or reimagined to show the important components without being so small.

Response:

Thanks Reviewer #1 for the constructive comment. We removed two less important panels in the Figure 6 and reimagined the figure.

Why do all images use a different color palette? Can this be fixed to appear as a cohesive set of data?

Response:

Thanks Reviewer #1 for the constructive comment. We re-plotted most of figures using a cohesive color palette.

These things do not require content change - but I would suggest that the authors would have better uptake of their manuscript with some attention to the appearance given the impact of the work.

Response:

We greatly appreciate Reviewer #1 for the constructive comments and suggestions. We have made the suggested modifications to the figures as described above.

Reviewer #1 (Remarks on code availability):

The data and database made available is excellent. All files were accessible.

Reviewer #2 (Remarks to the Author):

This is an interesting study comparing macaque sequence data from targeted sequencing of coding regions for a selection of human inherited retinal dystrophy genes (IRDs) and neurodevelopmental genes. Comparison of allele frequencies for macaque and human provides useful information for human variants, particularly those of uncertain significance. Additionally, one example of a compelling macaque model of an inherited optic nerve disease was presented. This is a well reasoned study using appropriate methods and with interesting results. The manuscript is also well written.

Response:

We highly appreciate the supportive and positive feedback provided by Reviewer #1.

Several comments for the authors to consider:

1) The primary limitation of this work is absence of phenotype for the sequenced individuals. The work would be more compelling and impactful if phenotype data was also available. Any plans to do this? Perhaps for the subcohort that underwent exome sequencing?

Response:

Thanks Reviewer #2 for the constructive comment. Given that most of the sequenced individuals lack available phenotype data, we have included this suggestion as a potential future direction in the discussion section:

“One potential enhancement is to integrate the phenotype data of the sequenced individuals into the mCED browser. This addition will facilitate the study of genotype-phenotype relationships and the development of animal models.”

2) For the RPGRIP1 variant it would be helpful for the reader to make clear that this gene is on

the X chromosome and is a cause of X-linked disease in humans. Also it would be interesting to know if the carriers of this variant are males or females. Similarly, for the PLA2G6 gene it would be helpful to note that this is a cause of autosomal recessive disease.

Response:

Thanks Reviewer #2 for the constructive suggestion. *RPGRIP1* gene is located on the chromosome 7 in the rhesus macaque genome and on the chromosome 14 in the human genome (please see the figures below. <https://www.ncbi.nlm.nih.gov/gene/57096>). Therefore, *RPGRIP1* is not a cause of X-linked disease in rhesus macaques or humans.

Indeed, *PLA2G6* is an autosomal recessive disease gene. To address Reviewer #2's comment, we edited the corresponding text in the manuscript to pinpoint disease inheritance patterns of the identified putative pathogenic variants as following.

“RPGRIP1, an autosomal recessive gene accounting for about 5% of Leber congenital amaurosis and also associated with cone-rod dystrophy and retinitis pigmentosa”

“PLA2G6, an autosomal recessive gene linked to Parkinson diseases and neurodegeneration”

“EYS, a common gene for retinitis pigmentosa in autosomal recessive inheritance mode”

“ABCA4, a major gene associated with Stargardt and other macular-related diseases in autosomal recessive inheritance mode”

3) The data for human low frequency variants of unknown significance that exhibit higher frequency in macaques is clearly useful to support a benign classification, and several examples are provided showing this. It would be helpful to know how many (or percent) of human VUS are in this category. This information may be included in the figures but it is a bit difficult to tease

out and as this is a significant point of the manuscript it would be helpful to have in text.

Response:

Thanks Reviewer #2 for the great suggestion. We added the following text in the manuscript:

“Collectively, 107 human variants labeled as 'Uncertain significance' and 52 human variants labeled as 'Conflicting_interpretations_of_pathogenicity' in ClinVar are rare in humans (human AF < 0.1% in gnomAD) but show a higher frequency in macaques (macaque AF > 3.8%), supporting a benign classification.”

4) In figure 2F what is included in the 'other' category? Synonymous variants? Intronic? Stop loss? UTR variants? As this appears to be the major category it would be helpful to know what is included.

Response:

Thanks Reviewer #2 for the great suggestion. In the “other” category, it includes the following variants: intron_variant (15354 rhesus macaque variants) , synonymous_SNV (17276 rhesus macaque variants), 3_prime_UTR_variant (509 rhesus macaque variants), downstream_gene_variant (1125 rhesus macaque variants), upstream_gene_variant (649 rhesus macaque variants), 5_prime_UTR_variant (353 rhesus macaque variants), coding_sequence_variant (3 rhesus macaque variants), start_lost (6 rhesus macaque variants), stop_retained_variant (9 rhesus macaque variants) and stop_lost (2 rhesus macaque variants). We added this information in the figure legend.

Reviewer #3 (Remarks to the Author):

In their manuscript entitled “Genetic diversity of 1,845 rhesus macaques improves genetic variation interpretation and identifies macaque models for inherited human retinal and neurodevelopment diseases,” Wang et al. generate a catalogue of variants using a targeted sequencing approach for 374 neurodevelopmental and retinal disease genes and present their findings. With the reverse genetic approach – screening of 1,845 of rhesus macaques from different primate research centers for single nucleotide variants (SNVs) – the authors identify more than 47,000 SNVs. In their comparison of the rhesus macaque variants to human genetic variation databases, the authors observe that common variants in rhesus macaque that are also present in human are likely polymorphic and that variants that are rare in both species are more likely disease associated. In addition, to identify some likely pathogenic variants, the authors investigate one disease (autosomal dominant optic atrophy) in more detail and show that rhesus macaque may serve as a suitable disease model. Furthermore, Wang et al. developed a machine learning-based prediction model and a searchable database. The intersection with human variants to decipher (likely) benign from pathogenic variants is a particular strength. The prediction model and searchable database will likely be of particular interest to the human genetics community.

The combination of approaches, tools, and databases included in the manuscript is unique, especially given that the database seems to include additional genes beyond the 374 neurodevelopmental and retinal disease genes. The different directions of the manuscript may attract a broader audience. At the same time, I feel that the transitions could be reemphasized

to make the manuscript more cohesive.

Response:

We are grateful for the encouraging and positive feedback by Reviewer #3.

I had some difficulty determining the genes included in the searchable database, as some genes (e.g., genes within the MHC locus) did not provide a match in the database. Wang et al. state that there is a larger/different dataset that will be subject to another publication. However, some greater detailed information about the database and its content may be useful for readers interested in utilizing the database. For example, could a gene list (or list of excluded genes) be provided?

Response:

Thanks Reviewer #3 for the great suggestion. We provided a downloadable list of genes that have genetic variants in the database at the <https://ird.research.bcm.edu/macaque/downloads.html>. This gene list is based on the data from both gene panel capture sequencing (presented in this paper) and whole exome sequencing (will be presented in another publication).

The genes within the MHC locus in rhesus macaques are more complex than those in human and exhibit copy number polymorphism¹. Additionally, the genes within the MHC locus in rhesus macaques may have different gene names from their homologous genes in humans, for example, MAMU-DOA, MAMU-DOB, MAMU-DMA, MAMU-DRB1, and MAMU-F.

Related to the database, would it be possible to provide the data for different human genome assemblies (hg38 and/or the T2T assembly) to provide access to a broader community?

Response:

Thanks Reviewer #3 for the constructive comment. We have identified the orthologous SNPs of the rhesus macaque genetic variants in the human genome hg38 assembly, and updated the database browser to include the relevant information.

Wang et al. did not discuss the origin of the rhesus macaque included in this study in their manuscript. More specifically, are all rhesus macaque housed in the primate centers and included in this study of Indian origin? Or were Chinese rhesus macaques also included? If Chinese rhesus macaque were included, did the different rhesus macaque population show varying allele frequency/disease allele distribution?

Response:

Thanks Reviewer #3 for the insightful comments. All rhesus macaques in this study are housed in the primate research centers. The U.S. research colonies of rhesus macaques were primarily established using animals imported from India decades ago, although a much smaller number of Chinese-origin rhesus macaques have been introduced to some colonies over time. Tracing the precise geographic origins of the U.S. research rhesus macaques in this study is not

feasible. However, we estimated them using principal component analysis (PCA) of their genotype data. We generated a PCA plot based on the genotype data from 1845 rhesus macaques along with data from three wild-caught Chinese rhesus macaques (shown in the following figure). The three wild-caught Chinese rhesus samples clustered together with a low value on PC1. Since the genotype data primarily cover the coding regions of 374 genes through targeted sequencing rather than the entire genome, the delineation of the Chinese cluster is not as clear. However, the PCA result indicates that the wild-caught Chinese samples cluster with a few samples from CNPRC and SNPRC. This result aligns with the knowledge that both CNPRC and SNPRC house a few known Chinese and Chinese/Indian hybrid animals. However, due to the limited number of predicted Chinese or Chinese/Indian hybrid animals in this study, it is not sufficient to accurately estimate allele frequency or disease allele distribution among them.

The genetic diversity within rhesus macaque populations in different primate centers can vary dramatically. Did the authors observe differences between different primate centers?

Response:

Thanks Reviewer #3 for the insightful question. To evaluate the variation in genetic diversity within rhesus macaque populations across different primate centers, we computed the allele frequencies of identified single nucleotide polymorphisms (SNPs) based on the allele count and allele number per primate center, respectively. Subsequently, we calculated the Pearson correlation of allele frequencies among these centers. Our analysis revealed a high correlation of allele frequencies across different primate centers, with Pearson correlation coefficients ranging from 0.93 to 0.99 (shown in the following heatmap plot of the correlation). This finding suggests minimal differences in genetic diversity among the centers. Notably, the correlation between Cayo Santiago (Caribbean Primate Research Center) and the other primate centers was relatively lower, with Pearson correlation coefficients ranging from 0.93 to 0.95. This result

aligns with the result in a prior paper that showed relatively lower heterozygosity in this colony (Fig. S17 in Warren et al's paper)². The reason for the different allele frequency in Cayo Santiago is either that the founding animals came from a different part of Indian, or that there was a population bottleneck on Cayo after the animals were introduced, or both.

Did some populations show homogenization/signs for inbreeding?

Response:

Thanks Reviewer #3 for the insightful question. We calculated method-of-moments F inbreeding coefficient for each individual based on the genotype data in this study and compared the distribution of F among different primate centers. As shown in the figure below, mean of F in rhesus macaque populations across different primate centers range from 0.010 to 0.075, similar to the estimation in a prior paper based on whole genome sequencing data of rhesus macaques (Fig. S18 in Warren et al's paper)². In contrast, the mean of F inbreeding coefficients in human is slightly higher than 0.1 (Figure S2 in Harris et al's paper)³. These results suggest inbreeding in rhesus colonies is not zero, but it is lower than in most human populations.

How did the authors account for variation between different primate centers in their model building?

Response:

Thanks Reviewer #3 for the insightful question. Given our focus on the impact of genetic variants on rhesus macaques in general rather than within specific primate centers, and considering the high similarity in genetic diversity across different centers, we incorporated the aggregated allele frequency across all eight primate centers into our model building.

Are the observed allele frequencies in this study aligned with allele frequency distribution in wild rhesus macaque populations? Related to this, is the number of pathogenic variants aligned with expectations based on population genetic analyses? Or did the authors see enrichment/depletion of pathogenic variants between primate centers compared to wild-caught rhesus macaques?

Response:

Thanks Reviewer #3 for the insightful questions.

The vast majority of the rhesus macaques analyzed in our study were Indian-origin rhesus macaques, and we know of no large or even moderate amount of allele frequency data from wild Indian-origin rhesus. Consequently, there is no available data that can answer what the allele frequency of pathogenic variants is in the appropriate wild macaque populations. Therefore, we cannot directly compare our data to the wild animals.

To assess whether the number of our identified pathogenic variants is reasonable, we compared them with the results from a prior paper (Warren et al Science 370: eabc6617)². In that prior paper, a total of 20,400 Likely Gene Disruptive (LGD) SNVs from 21,120 protein-coding genes were identified across 853 captive rhesus macaques, averaging 0.97 LGD SNVs per gene. In our dataset, we identified 77 LGD SNVs from 374 protein-coding genes across 1845 captive rhesus macaques, averaging 0.21 LGD SNVs per gene. (Please note the definition of LGD SNVs in the prior paper differs from the LoF and pathogenic SNVs in our manuscript, thus the number of LGD SNVs, i.e. 77, is also different from those listed in our manuscript). Since the 374 genes are associated with inherited retinal diseases and neurodevelopmental

disorders, we believe that they are under stronger natural selection and constraint than the “average” rhesus macaque gene. Therefore, the lower number of LGD SNVs observed in genes related to the inherited diseases compared to the genome-wide average is consistent with expectation.

Regarding the distribution of the identified pathogenic variants among primate centers, most of the pathogenic variants have low allele frequencies and are present in only a single primate center. Overall, the number of pathogenic variants identified in each primate center is weakly correlated with the number of rhesus macaques sequenced from each primate center (correlation coefficient=0.64, $p=0.09$).

In their abstract, Wang et al. state the identification of more than 47,000 high quality SNV. The legend of Figure 2b indicates 46,444 variants overall. What accounts for the discrepancy?

Response:

Thanks Reviewer #3 for the comment. As indicated in the legend of Figure 2b “Histogram showing the allele frequency distribution of the **autosomal** SNVs in each primate center and in all the centers (number of the SNVs = 46,444)”, this plot only includes autosomal SNVs and did not include SNVs on the X chromosome. Indeed, we identified 47,743 SNVs considering genetic variants on both the autosomes and the X chromosome.

The scale of the Y axis in Figure 2b appears to be off. I find it difficult to determine the actual allele frequency of the different variants on the X-axis. For example, where is .25 exactly located on the X axis? Why is there a white space within the first bar (0.00)? To what does all refer?

Response:

Thanks Reviewer #3 for the constructive comment. As majority of the SNVs are rare in the rhesus macaque population, the first bar is much higher than the remaining bars in the Histogram. Therefore, we previously truncated the interval between 2,500 and 50,000 in the Y axis. To address Reviewer’s comment, we have modified Figure 2b to show the full spectrum of the Y axis and added minor grid to the figure.

Is there a reason why the colors used for the different primate centers are not kept the same between Figure 2a and 2b?

Response:

Thanks Reviewer #3 for the constructive comment. We modified Figure 2a and Figure 2b to match the colors of the primate centers between the two figures.

The number of pathogenic variants in rhesus macaque (Figure 3) seems rather limited. Would it be possible to include a larger dataset (more genes/variants) to further support the findings?

Response:

Thanks Reviewer #3 for the great suggestion. We aggregated the pathogenic variants from three human disease variant databases and also include the orthologous pathogenic human variants that have the genomic regions sequenced in our data but with an allele frequency of 0 in the rhesus macaque population (“Filtering of benign variants” in Methods). As a result, we have now collected 11,208 pathogenic variants. However, the general patterns of allele frequency distribution in the new dataset remain consistent with our previous results.

A deeper engagement with the putative deleterious variants may be of interest to a broader research community.

Response:

Thanks Reviewer #3 for the great suggestion. We compared the distribution of established variant prediction scores for putative deleterious variants predicted by our HM score with those of the remaining variants. Consequently, these putative pathogenic variants consistently exhibited higher established variant scores than the remaining variants (shown in Supplementary Fig S5).

Additionally, we investigated the predicted putative deleterious variants and included the following sentences to the corresponding section:

“Consequently, we identified 68 putative deleterious missense variants in 49 IRD genes and 8 ND genes from 274 macaques. Notably, some of the identified putative deleterious variants were found in haploinsufficiency disease genes in autosomal dominant inheritance mode, such as *ITM2B*, *JAG1*, *KIF11*, *LRP5*, *NR2F1*, *PRPF31*, *SNRNP200*, *VCAN*. This suggests that the rhesus macaque carriers of the identified putative deleterious variants can be directly examined to confirm the phenotype. Additionally, some genes in autosomal recessive inheritance have multiple putative deleterious variants identified, for example, *CLN3*, *CNNM4*, *ABCA4*, *CC2D2A*, *EYS*. This provides potential breeding candidates to develop rhesus macaque disease models. Overall, these results offer priority for future analyses and the development of new rhesus macaque models of diseases.”

Overall, I found figure 6 less informative and essential for the manuscript.

Response:

Thanks Reviewer #3 for the constructive comment. Figure 6 illustrated the utilization of the variants and genotype data in the mCED browser, which serves as a unique and valuable resource for the research community, greatly enhancing the significance of this manuscript. We removed two less important panels in the previous Figure 6 and reimagined the figure to enhance readability.

Reference:

1. Doxiadis, G. G. M., de Groot, N., Otting, N., Blokhuis, J. H. & Bontrop, R. E. Genomic plasticity of the MHC class I A region in rhesus macaques: extensive haplotype diversity at the population level as revealed by microsatellites. *Immunogenetics* **63**, 73–83 (2011).
2. Warren, W. C. *et al.* Sequence diversity analyses of an improved rhesus macaque genome enhance its biomedical utility. *Science* **370**, (2020).
3. Harris, R. A. *et al.* Whole Genome Analysis of SNV and Indel Polymorphism in Common Marmosets (*Callithrix jacchus*). *Genes (Basel)* **14**, 2185 (2023).

REVIEWERS' COMMENTS

Reviewer #1 (Remarks to the Author):

The authors have successfully addressed all of my comments and I believe the manuscript is acceptable for publication in this form.

Reviewer #1 (Remarks on code availability):

this is appropriate and does not require any further review

Reviewer #2 (Remarks to the Author):

In this revised manuscript the authors have satisfactorily addressed the concerns raised by the review.

Reviewer #3 (Remarks to the Author):

I appreciate the detailed response to my comments and questions by Wang et al. Overall, the authors addressed my concerns satisfactorily. For readers of the manuscript, it may be informative to provide some of the details shared in the rebuttal in the manuscript (e.g., a brief mention of the composition of the rhesus macaque populations in the methods section or a supplemental information).

In addition, I have two minor, fully discretionary comments:

In Figure 2, I find it not easy to distinguish light pink and grey in the section where most individuals cluster.

I have a minor comment with regard to Figure 5. People with altered color vision may have difficulty distinguishing between green and red.